

# Influence of periodontal treatment on blood microbiotas: a clinical trial

Wenyi Zhang[1], Yang Meng[2], Jin Jing[1], Yingtao Wu[1] and Shu Li[3]

[1] Department of Periodontology, Qingdao Stomatological Hospital Affliated to Qingdao University, Qingdao, Shandong, China
[2] Department of Prosthodontics, Qingdao Stomatological Hospital Affliated to Qingdao University, Qingdao, Shandong, China
[3] Department of Periodontology, School and Hospital of Stomatology, Cheeloo College of Medicine, Shandong University & Shandong Key Laboratory of Oral Tissue Regeneration & Shandong Engineering Laboratory for Dental Materials and Oral Tissue Regeneration, Jinan, Shandong, China

## ABSTRACT

**Objective**. To investigate the effects of periodontal treatment on the abundance and diversity of blood microbiota.

**Methods and Materials**. Twenty-seven periodontitis patients were randomly allocated to a control group (A) and two test groups (B1 and B2). Group A patients received full-mouth scaling and root planing (SRP), group B1 patients received subgingival glycine air polishing (GAP) right after SRP, and group B2 patients received subgingival glycine air polishing right before SRP. Peripheral blood samples were obtained at the baseline, the day after periodontal treatment, and 6 weeks after treatment and evaluated using nested polymerase chain reaction and 16SrRNA Gene Sequencing (Miseq platform).

**Results**. All participants exhibited significant improvements in the clinical parameters evaluated at the 6-week follow-up visit compared to the values at the baseline, but no significant differences were observed between the three groups. The total bacterial count was lowest in group B2. The bacterial species diversity ($\alpha$-diversity) in group B1 was significantly higher (Chao-1 index, $P = 0.03$) and *Porphyromonas* and *Pantoea* were the dominant genera (linear discriminant analysis (LDA > 2)) in this group the day after treatment compared to the baseline. No significant difference was detected in the relative abundance and $\alpha$-diversity of blood microbiota between the baseline and 6 weeks after treatment.

**Conclusion**. Local periodontal treatment merely disrupts the stability of blood microbiota in the short term. Periodontitis treatment using full-mouth SRP followed by adjunctive GAP is a promising approach to reduce the introduction of bacteria into the bloodstream during the procedure.

Corresponding authors
Yingtao Wu, 763466324@qq.com
Shu Li, lishu@sdu.edu.cn

## INTRODUCTION

Periodontitis is caused by the accumulation of oral microorganisms in dental plaque and is characterized by an irreversible breakdown of the periodontal tissues due to complex interactions between chronic bacterial infections and the inflammatory host response (*Dhotre et al., 2018*). In addition to the localized inflammatory response, periodontal diseases may affect systemic health (*Scannapieco, Bush & Paju, 2003*). In individuals

with periodontitis, the inflamed ulcerated crevicular or pocket epithelium around the teeth can facilitate the translocation of oral microorganisms from the oral cavity to the bloodstream (*Marsh, 2005*; *Parahitiyawa et al., 2009*). Emerging evidence suggests that the blood microbiota plays a role in the development of periodontitis and systemic diseases, such as cardiovascular diseases (*Tonetti et al., 2007*), diabetes (*Mattila et al., 1989*), Parkinson's disease (*Kumar, 2017*), and infective endocarditis (*Dhotre et al., 2018*).

The development of periodontitis may be attributable to the global balance of the microbial flora rather than the presence of specific periodontal pathogens (*Camelo-Castillo et al., 2015*). Conventional periodontal treatment, which involves the mechanical removal of supra and subgingival bacterial deposits, primarily by scaling and root planing (SRP) decreases the inflammatory response and enhances healing (*Buhlin et al., 2009*). Additionally, open-ended 16SrRNA gene sequence analysis results suggest that periodontal treatment, primarily SRP, can alter the microbiota composition in salivary and subgingival plaque (*Liu et al., 2018*; *Belstrøm et al., 2018*). Though several studies have reported the detection of periodontal pathogens in blood using culture (*Marín et al., 2016*; *Lafaurie et al., 2007*) or molecular (*Castillo et al., 2011*; *Ambrosio et al., 2018*) techniques, these studies focused on specific periodontal pathogens within a short period. Published literature describing the influence of periodontal treatment on whole blood microbiota in the short and long term is sparse.

SRP may act as a potent inflammatory stimulus immediately after treatment, inducing an acute systemic inflammatory response (*Morozumi et al., 2018*; *Graziani et al., 2015*; *Graziani et al., 2010a*; *Graziani et al., 2010b*), and is associated with postoperative bacteremia (*Moëne et al., 2010*). Several studies have demonstrated that glycine air polishing (GAP) has a high plaque removal efficiency. The only reported side effects of GAP are rare incidence of air emphysema (*Petersilka, 2011*; *Lu et al., 2018*)). GAP has been recommended as a maintenance treatment and an adjunct to SRP for individuals with untreated periodontitis (*Tsang, Corbet & Jin, 2018*; *Caygur et al., 2017*). However, whether GAP leads to the introduction of more or less oral bacteria into the bloodstream remains unclear and thus, further investigations are needed. Therefore, the aim of the present study was to investigate the effects of periodontal treatment on the abundance and diversity of blood microbiota. In addition, we used 16SrRNA gene sequencing to assess the blood microbiota following SRP and adjunctive GAP treatment of periodontitis patients.

## METHODS AND MATERIALS

### Study design and participants

The present single-blinded randomized controlled trial (RCT) was reviewed and approved by the Research Ethics Committee at the Shandong Stomatological Hospital, China (Ref No: 20191003). The RCT was conducted in accordance with the Helsinki Declaration of 1975 (revised in 2013) and registered in the Clinical Trial Registry (http://www.chictr.org.cn, ChiCTR1900021599). The study participants were recruited from the Department of Periodontology at Qingdao Stomatological Hospital from September 2018 to May 2019 according to the predefined selection criteria (Table 1). An informed written consent form was obtained from all participants.

**Table 1  Patient selection criteria.**

| Inclusion criteria | Exclusion criteria |
|---|---|
| I. Patients with stage II, III, or IV periodontitis according to the criteria defined in the 2018 World Workshop on the Classification of Periodontal Diseases and Conditions[23]. | I. Pregnant or lactating women V. Smoking |
| II. Age: 20 to 70 years old | II. Received systemic antibiotics or immunosuppressants within the previous 6 months. |
| III. Presence of a minimum of 20 teeth with no incompatible dentition such as orthodontic bands or partial dentures | III. Unsuitable for extensive peri-odontal treatment. |
| IV. Untreated periodontitis or only received supragingival scaling more than 1 year earlier. | IV. History of any systemic disease related to periodontitis |

Fifty-six patients were initially evaluated for eligibility and 29 patients were excluded for various reasons. Twenty-seven patients diagnosed with generalized stage II, III, and IV periodontitis (*Papapanou et al., 2018*) were enrolled in this three-arm RCT (Fig. 1). Assessment of the sample size was based on power calculation for a two-tailed comparison (G*Power 3.1) (*Faul et al., 2007*; *Wennström, Dahlén & Ramberg, 2011*). The results indicated that the inclusion of 27 patients (considering a 15% drop-out risk) would allow for the detection of a mean difference of 0.5 mm in probing depth (PD) reduction between treatments, with 80% power and 5% alpha error.

## Process, randomization, and interventions

For each participant, baseline data collection, periodontal charting, and blood sample collection were performed at the first clinical visit before randomization and treatment (Fig. 1). Participants were randomized into a control group (Group A; $n = 9$) and two experimental groups (Groups B1 and B2; $n = 10$ for each group) with a random number table. Patients in the control group received supragingival ultrasonic scaling at the baseline visit, followed by subgingival full-mouth SRP 1 week later. The first intervention group (Group B1) received plaque staining, supragingival GAP, and supragingival ultrasonic scaling at the baseline visit, followed by full-mouth SRP and subgingival GAP at the 1-week visit. Participants in the second intervention group (Group B2) received the same therapy as the Group B1 participants; however, subgingival GAP was performed right before full-mouth SRP at the 1-week visit. Subgingival GAP was performed at sites with PD $\geq$ four mm (Fig. 2).

Supragingival and subgingival air polishing were performed using 65 $\mu$m and 25 $\mu$m glycine amino acid powder, respectively (Air-Flow Polishing Soft Powder; EMS, Nyon, Switzerland). For subgingival treatment, a specially designed nozzle (Perio-Flow, Nozzle, EMS, Switzerland) was inserted into the periodontal pocket. All periodontal procedures including full-mouth SRP were performed by the same periodontist using manual and ultrasonic instruments.
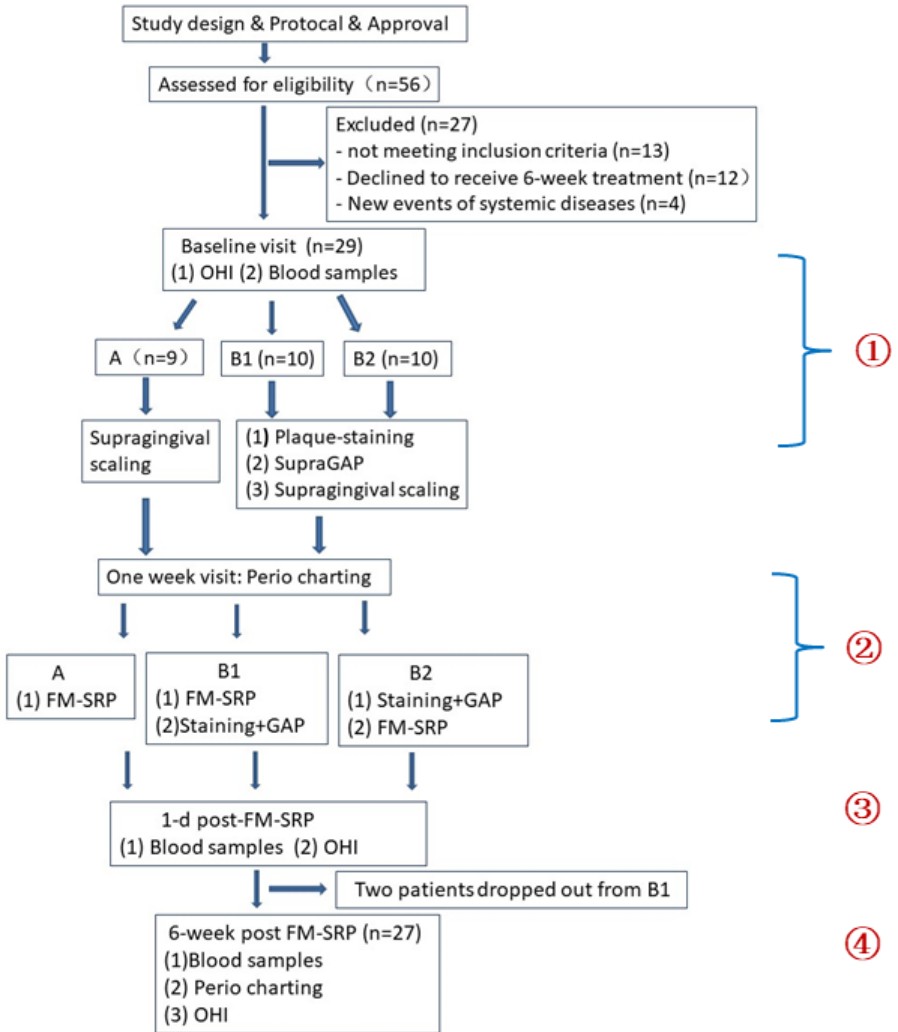

**Figure 1** **Flow chart representing the characteristics of the study groups and the study design.** FM-SRP: full-mouth scaling and root planing, GAP: glycine air polishing, OHI: oral health instruction.

## Periodontal assessment

Clinical periodontal examinations were performed two times——-at the one-week and 6-week follow-up visits after supragingival scaling to collect the following data: (1) PD (*Lu et al., 2018*), plaque index (PLI) (*Silness & Loe, 1964*), bleeding index (BI) (*Lu et al., 2018*), and bleeding on probing (BOP) (*Bundidpun, Srisuwantha & Laosrisin, 2018*). The examining periodontist was blinded to the patient assignments.

## Sample collection and DNA extraction

Peripheral blood samples were obtained by venipuncture in the antecubital fossa before treatment at the baseline as well as the next-day and 6-week follow-up visits. Before sampling, the skin overlying the vein was disinfected using 75% ethyl alcohol and 2% chlorhexidine to minimize skin contaminants. Each sample (3 ml whole blood) was stored

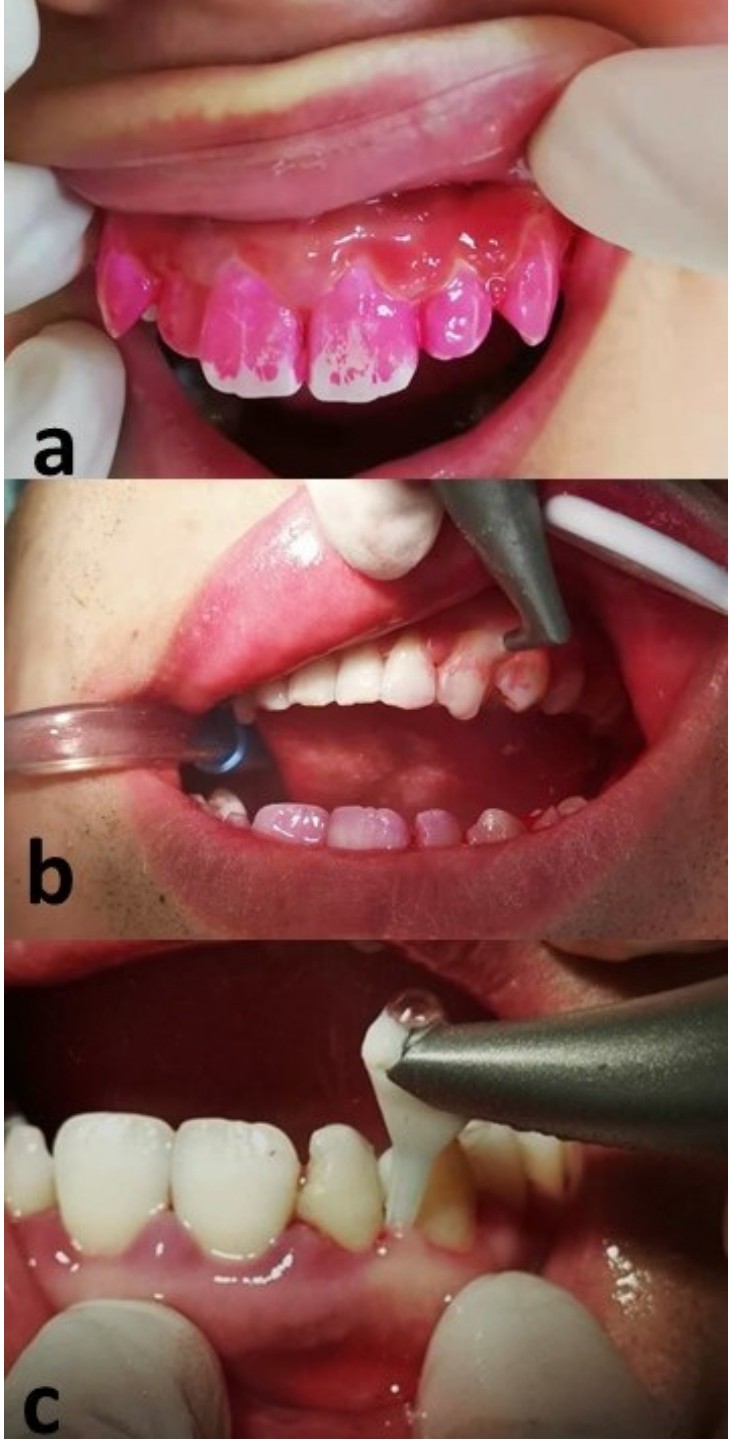

**Figure 2** **Description of glycine air polishing (GAP).** (A) Plaque staining before the treatment; (B) Removal of visible plaque using supragingival GAP; (C) Subgingival GAP performed with a plastic nozzle inserted into periodontal pockets.

at −80 °C immediately after collection until laboratory analysis. Bacterial genomic DNA was extracted from 200 ul of whole blood using a commercial kit (TianGen, TIANamp, China). The concentration and purity of DNA were assessed with the Micro Drop Ultra Micro Spectrophotometer (Bio-DL, TX, USA). Double distilled water was used as the negative control during the extraction process.

Nested polymerase chain reaction (nested PCR) amplification and 16SrRNA Gene Sequencing. Universal bacterial primers (27F 5′AGAGTTTGATCCTGGCTCAG-3′1492R5′-TACGGYTACCTTGTTACGACTT-3′) (*Fredriksson, Hermansson & Wilén, 2013*) were used to detect bacterial DNA in the blood samples. PCR amplification was carried out in a 25 µl reaction mixture containing 2× Taq Plus Master Mix (KAPA 2G Robust HotStart ReadyMix, Sigma-Aldrich, USA), 5 µM of the forward and reverse primers, 6 ng bovine serum albumin (BSA), and 30 ng of template DNA. A 2 kb DNA ladder was used as a size marker to assist the analysis of the PCR products. Subsequently, nested PCR was performed using the universal primers (336F 5′-GTACTCCTACGGGAGGCAGCA-3′ 806R 5′-GTGGACTACHVGGGTWTCTAAT-3′) (*Wei et al., 2016*) targeting the V3-V4 region of the 16SrRNA gene. Doubled distilled water was also used as the negative control.

The final PCR products were separated by 2% agarose gel electrophoresis. The ∼470 bp fragments obtained were purified using a QIAquick Gel Extraction Kit (Qiagen, Hilden, Germany) and sequenced on a MiSeq platform (Allwegene, China).

## Statistical analysis

Descriptive analyses of the quantitative variables were performed and the results are presented as the mean ± standard deviation (SD). The median, minimum, and maximum values were calculated for the BI and PLI. All parameters were tested for normality using the Shapiro–Wilk method. Repeated measures analysis of variance (ANOVA) was used to compare longitudinal changes in the patients in the three groups with a normal distribution. If the parameters did not follow a normal distribution, the Friedman non-parametric test was performed. All the statistical analyses were performed using the SPSS ver. 24.0 software (IBM Japan, Tokyo, Japan).

Sequences were analyzed using the pipeline tools QIME v 1.8.0 and Mothur v 1.34.4. Barcodes and primers were trimmed and raw reads with a quality value <20 were removed. The Mann–Whitney U-test was used to compare the bacterial species diversity (α-diversity; measured with the Chao-1 index, observed_species, PD_whole_tree, and Shannon index). Principal component analysis (PCA) based on the community structure (β-diversity) was conducted according to the distance matrix. Analysis of similarities (ANOSIM) was performed to compare the intra- and inter-group similarities based on the UniFrac distance. Microorganism features to distinguish the blood microbiotas specific to periodontitis before and after treatment were identified using a linear discriminant analysis (LDA) effect size (LEfSe) threshold of 2.0. Metastats analysis was performed to compare the relative abundance of bacteria pre- and post-treatment. All tests were two-tailed. A *p*-value <0.05 was considered to indicate statistical significance.

**Table 2** Baseline characteristic of the study participants.

| Variables | Group A (n = 9) | Group B1 (n = 8) | Group B2 (n = 10) |
|---|---|---|---|
| Age (mean ± SD) | 38.7 ± 7.4 | 47.5 ± 10.9 | 43.6 ± 11.7 |
| Female sex (n, %) | 4 (44.4%) | 3 (37.5%) | 4 (40%) |
| Stage (n/%) | | | |
| Stage II | 2 (22.2%) | 2 (25%) | 3 (30%) |
| Stage III | 6 (66.7%) | 5 (62.5%) | 6 (60%) |
| Stage IV | 1 (11.1%) | 1 (12.5%) | 1 (10%) |

**Notes.**
Variables expressed as mean standard deviation (SD) or n (%).

## RESULTS

### Clinical data

Of the 29 patients included in the study, 27 completed the 8-week trial (Table 2). Two participants from Group B1 dropped out due to business conflicts. The ages of the 27 patients ranged from 28 to 61 years (mean age, 41 years). Over half (55.6%) of the patients were male and most patients (63%) were diagnosed with stage III periodontitis. There were no significant differences in the study groups in terms of the participants' age, gender, and periodontitis stage.

Most of the clinical parameters indicated significant improvements in the three groups after treatment ($P < 0.05$). The percentage of BOP-positive, PD, and BI = 3 and 4 sites decreased sharply and the percentage of BI = 1 and 2 sites increased significantly at the 6-week visit after treatment (Table 3). However, there was no statistically significant difference among the groups in all clinical parameters between the baseline and 6 weeks after full-mouth SRP.

Nine patients (33.3%) complained about systemic symptoms the day after the full-mouth SRP. These symptoms consisted of chills, low grade fever (no higher than 38 °C), headache, fatigue, and drowsiness, which persisted for no longer than 24 h before disappearing completely.

### Semiquantitative analysis and general information for 16SrRNA gene sequencing

Eight participants were prescribed antibiotics due to the common cold within 1 week prior to the 6-week follow-up visit and their blood samples from the baseline and the day after treatment were analyzed. The total bacteria in 75 blood samples were evaluated using nested PCR (Fig. 3). Positive PCR gel strips indicated the presence of the 16SrRNA gene in the blood and brighter strips indicated a higher amount. The brightness levels of the gel strip are summarized and the darkest strips were found in Group B2 the day following the full-mouth SRP. Additionally, 70% of the gel strips were obviously less bright 1 day post-treatment (Table 4). The negative control (double distilled water) yielded no PCR products (Fig. 3).

A total of 37 blood samples were selected (12 from Group A, 11 from Group B1, 14 from Group B2) for further high-throughput 16SrRNA gene sequencing. A total of 1,252,175

**Table 3 Clinical parameters of individuals pre- and post-treatment ($n = 27$).**

| Clinical Parameters | Group A ($n = 9$) | Group B1 ($n = 8$) | Group B2 ($n = 10$) |
|---|---|---|---|
| PD (mm) | | | |
| Pre-treatment | $4.0 \pm 0.49$ | $4.28 \pm 0.67$ | $3.92 \pm 0.54$ |
| Post-treatment | $3.25 \pm 0.23$ | $3.21 \pm 0.24$ | $3.23 \pm 0.24$ |
| *P* value | 0.003 | 0.005 | <0.001 |
| BOP (%) | | | |
| Pre-treatment | $81.3 \pm 10.8$ | $86.4 \pm 15.1$ | $82.2 \pm 13.1$ |
| Post-treatment | $45.6 \pm 14.6$ | $41.93 \pm 13.1$ | $42.2 \pm 9.5$ |
| *P* value | <0.001 | 0.005 | <0.001 |
| BI = 1 (%) | | | |
| Pre-treatment | $2.2 \pm 2.3$ | $4.0 \pm 8.2$ | $5.7 \pm 6.7$ |
| Post-treatment | $25.8 \pm 10.3$ | $30.5 \pm 10.2$ | $27.7 \pm 9.3$ |
| *P* value | 0.003 | 0.005 | 0.002 |
| BI = 2 (%) | | | |
| Pre-treatment | $53.2 \pm 17.5$ | $35.8 \pm 14.3$ | $46.6 \pm 16.8$ |
| Post-treatment | $63.7 \pm 10.5$ | $59.8 \pm 8.4$ | $61.4 \pm 8.0$ |
| *P* value | 0.15 | 0.005 | 0.013 |
| BI = 3 (%) | | | |
| Pre-treatment | $31.6 \pm 10.1$ | $39.8 \pm 13.4$ | $33.9 \pm 13.2$ |
| Post-treatment | $9.6 \pm 12.2$ | $9.6 \pm 7.8$ | $10.7 \pm 7.5$ |
| *P* value | 0.02 | 0.001 | 0.002 |
| BI = 4 (%) | | | |
| Pre-treatment | $13.0 \pm 11.7$ | $20.4 \pm 19.3$ | $13.7 \pm 12.1$ |
| Post-treatment | $0.9 \pm 2.7$ | 0 | $0.2 \pm 0.7$ |
| *P* value | 0.005 | 0.005 | 0.002 |
| PLI = 0 (%) | | | |
| Pre-treatment | $3.6 \pm 6.2$ | $1.6 \pm 2.6$ | $6.3 \pm 11.7$ |
| Post-treatment | $13.9 \pm 18.6$ | $5.5 \pm 11.2$ | $9.2 \pm 17.6$ |
| *P* value | 0.025 | 0.655 | 0.157 |
| PLI = 1 (%) | | | |
| Pre-treatment | $50.3 \pm 24$ | $50.2 \pm 34.1$ | $58.2 \pm 23.2$ |
| Post-treatment | $46.9 \pm 20.1$ | $40.1 \pm 22.6$ | $66.2 \pm 26.7$ |
| *P* value | 0.665 | 0.476 | 0.431 |
| PLI = 2 (%) | | | |
| Pre-treatment | $36.7 \pm 19$ | $30.4 \pm 15.8$ | $26.3 \pm 17.5$ |
| Post-treatment | $26.5 \pm 27.6$ | $37.7 \pm 19.5$ | $20.0 \pm 18.1$ |
| *P* value | 0.267 | 0.457 | 0.295 |
| PLI = 3 (%) | | | |
| Pre-treatment | $9.8 \pm 10.0$ | $17.8 \pm 23$ | $9.2 \pm 13.3$ |
| Post-treatment | $15.3 \pm 17.9$ | $16.7 \pm 19.1$ | $4.6 \pm 10.7$ |
| *P* value | 0.705 | 0.48 | 0.655 |

**Notes.**

Values mean ± are the standard deviations.

PD, periodontal depth; BOP, bleeding on probing; BI, bleeding index; PLI, plaque index.

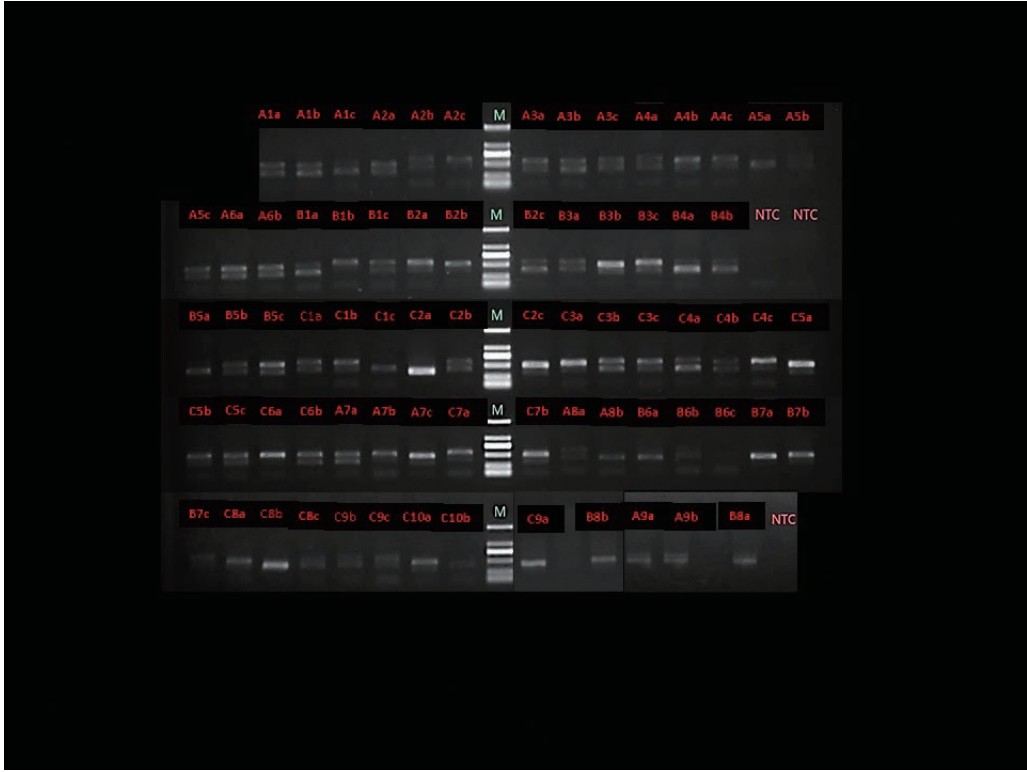

**Figure 3  Agarose gel electrophoresis of representative polymerase chain reaction amplification result with total bacterial 16S rRNA general primers.** M - Marker, NTC - negative control (double distilled water as the DNA template). A - Group A, B - Group B1, C - Group B2, 1 - sample number, a - baseline, b - the day after full-mouth SRP, c - 6 weeks after full-mouth SRP.

**Table 4  Semi-quantitative analysis of total bacteria in blood samples using nested PCR.**

| Strip brightness | Group A | Group B1 | Group B2 |
|---|---|---|---|
| [*]Obviously weaker | | | |
| 1 d after treatment | 22% | 25% | 70% |
| 6 weeks after treatment | 16.7% | 33.3% | 57.1% |
| [**]Marked enhancement | | | |
| 1 d after treatment | 22% | 12.5% | 10% |
| 6 weeks after treatment | 0 | 33.3% | 14.3% |
| [***]Unchanged | | | |
| 1 d after treatment | 56% | 62.5% | 20% |
| 6 weeks after treatment | 83.3% | 33.3% | 28.6% |

**Notes.**
[*]strip brightness is obviously weaker compared to the baseline.
[**]strip brightness is markedly enhanced compared to the baseline.
[***]strip brightness has no change compared to the baseline.

unique sequences and 383 operational taxonomic units were obtained, with a mean value of 27,826 clean tags/sample. In total, there were 14 phyla, 32 classes, 54 orders, 97 families, 171 genera, and 41 species taxa in these blood samples.

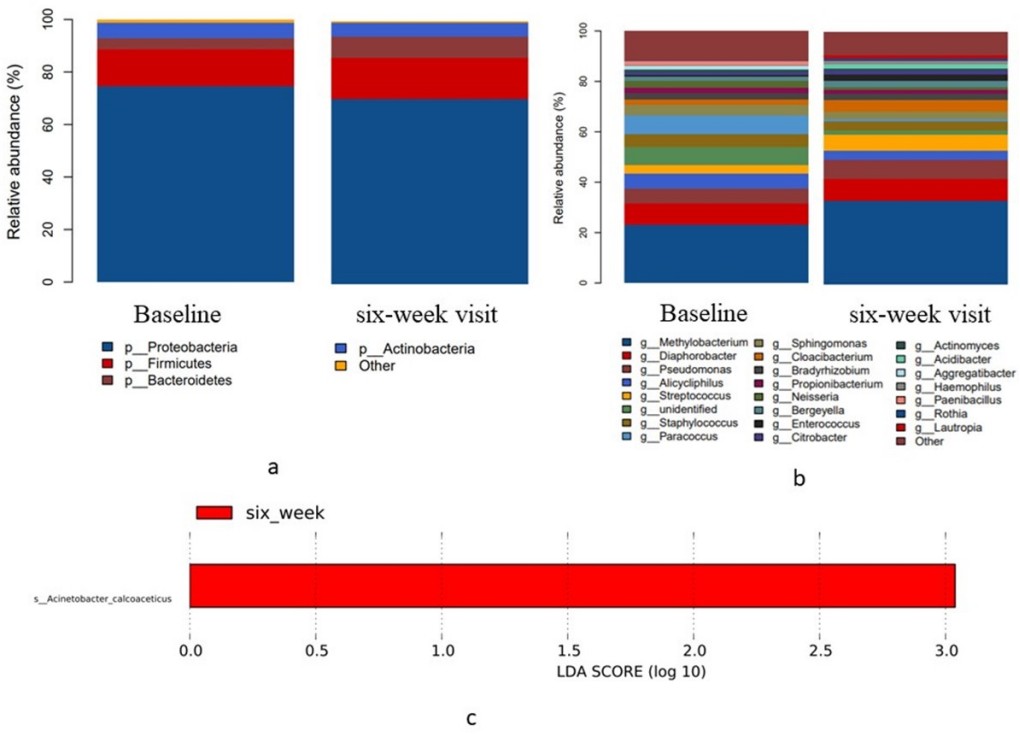

**Figure 4** **Taxonomic differences in the microbial 16S rRNA gene in the blood before treatment (baseline) and 6 weeks post-treatment (6-week visit).** (A) Relative abundance of the four most common bacterial phyla (B) Relative abundance of the 23 most common bacterial genera (C) LEfSe analysis showed the dominant blood bacteria at 6 weeks after full-mouth SRP.

## Variations in blood microbiota at 6 weeks after periodontal treatment

The present study revealed that 99% of the bacteria in human blood are *Proteobacteria*, *Firmicutes*, *Bacteroidetes*, and *Actinobacteria* at the phylum level (Fig. 4A). The 23 most common genera at the baseline and 6 weeks after treatment are shown in Fig. 4B. The *Methylobacterium* genus had the highest relative abundance in blood relative to periodontitis. A comparison of the relative abundance, α-diversity, and β-diversity at the baseline and 6 weeks after treatment with a threshold filter of 1% revealed no statistical differences in the blood microbiota in the experimental groups. LEfSe analysis indicated that *Acinetobacter calcoaceticus* was the dominant species 6 weeks after treatment (Fig. 4C). These findings suggest there was no difference in the richness and community structure of blood microbiota at 6 weeks following the local periodontal treatment compared to the baseline levels.

## Impact of treatment on the blood microbiota the day after treatment

The day after full-mouth SRP, groups A (conventional full-mouth SRP) and B2 (subgingival GAP right before full-mouth SRP) showed no significant differences in the α-diversity and β-diversity compared to the baseline. On the contrary, Group B1 (subgingival GAP right after full-mouth SRP) exhibited higher α-diversity the day after full-mouth SRP

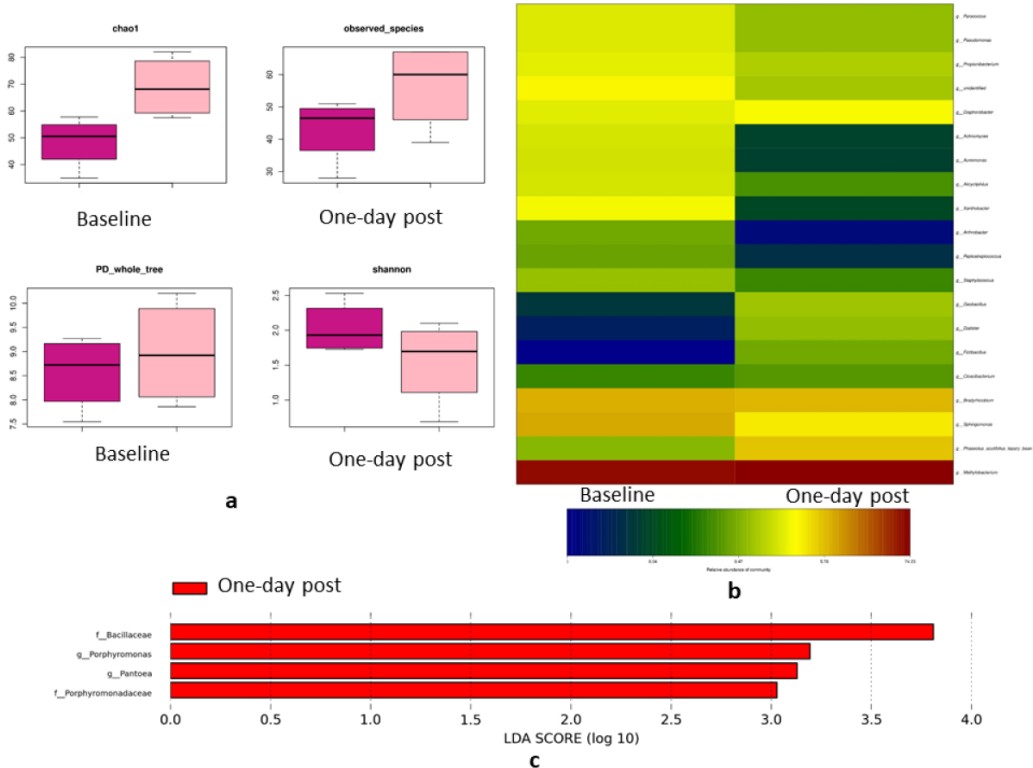

**Figure 5  Blood bacterial changes the day after treatment in Group B1.** (A) Indices of alpha-diversity in Group B1; Box plots depicting bacterial diversity according to the Chao-1 index, observed species, PD_whole_tree, and Shannon index. The black line in each column represents the median value. (B) Relative abundance of bacterial genus pre- and post-treatment. (C) The dominant bacteria in the blood the day after full-month SRP.

than at the baseline (Chao-1 index, $P = 0.03$) (Fig. 5A). The relative abundance of the *Allicycliphilus* genus decreased 13-fold (0.47% vs. 6.1%, $P = 0.019$) and six-fold (1.5% vs. 0.25% $P = 0.028$) in Groups A and B1, respectively. The *Methylobacterium* genus increased nine-fold (1.4% vs. 4.9%, $P = 0.049$) and the *Streptococcus-pneumoniae* species decreased 47-fold (7% vs. 0.3%, $P = 0.002$) in Group B2 the day after full-mouth SRP compared to the baseline values (Fig. 5B). LEfSe analysis indicated that *Porphyromonas* and *Pantoea* were the dominant genera in Group B1 the day after full-mouth SRP relative to the baseline (Fig. 5C). These findings suggest that periodontal treatment can alter blood microbiota diversity and relative abundance the day following local periodontal treatment.

## DISCUSSION

The effects of periodontal treatment (SRP) on the abundance and diversity of blood microbiota were investigated in the present RCT. For this purpose, peripheral blood samples from patients were periodically analyzed to identify bacterial genes using nested PCR targeting the 16SrRNA gene. The results of the study demonstrated that periodontal treatment for patients with untreated periodontitis altered the relative abundance and

diversity of blood microbiotas in the short term, but not in the long term. Subgingival GAP as an adjunct to full-mouth SRP had various effects on the blood microbiota the next day, leading to the recommendation of GAP for certain clinical applications.

Human blood contains the bacterial 16SrRNA gene (*Nikkari et al., 2001*; *Païssé et al., 2016*); however, more evidence is needed to rule out contaminants from the skin following phlebotomy, the manipulation of blood samples (*Wang et al., 2015*; *Ono et al., 2005*; *Bellot et al., 2010*), PCR reagents, and the sequencing pipeline (*Nikkari et al., 2001*). The negative controls used for the DNA extraction and nested PCR assays showed that the background was very low in comparison to the values obtained for the blood samples (Fig. 3). Moreover, the changes observed in the overall relative abundance and diversity of the microbiome in the blood eliminated the possibility that the microbiome variations were caused by contamination.

At the genus level, it is generally accepted that the most common oral bacteria include *Streptococcus, Haemophilus, Neisseria, Veillonella,* and *Prevotella* (*Pereira et al., 2017*; *Koren et al., 2011*), which comprise a small proportion of the blood microbiota. In the present study, the dominant bacterial genera in the blood samples evaluated were *Methylobacterium, Diaphorobacter, Pseudomonas, Cirtrobacter,* and *Cloacibacterium,* which are not the dominant genera in gut microbiota. In addition, the major components of the blood microbiota differ substantially from the microbiota detected in feces (*Qian et al., 2019*; *Lelouvier et al., 2016*). Taken together, the results of the present study do not support the hypothesis that the blood microbiota originates from the gut or oral cavity. However, the biological significance and the association of certain genera with a specific disease remain unclear.

The present study uncovered a stable bacterial community in the blood that was neither transient nor instantaneous. The diversity and relative abundance of bacteria may change at the next day after treatment but rebound to the same level as the baseline at the 6-week follow-up visit. Lafaurie (*Lafaurie et al., 2007*) reported that the majority of patients (81%) tested positive for bacteria in the bloodstream immediately after SRP, but only about 19% remained positive 30 min after the procedure. Similarly, periodontal probing resulted in bacteremia within a few seconds (*Bellot et al., 2010*). Oral microbes such as *Porphyromonas gingivalis, Aggregatibacter actinomycetemcomitans,* and *Fusobacterium nucleatum* have been detected immediately after tooth brushing and are considered bacteremia (*Marín et al., 2016*). Bacteremia is a transient process that occurs when bacteria enter the bloodstream in the pathological condition. Hence, the conventional definition of bacteremia is based on the complete sterility of blood under a healthy condition.

Our results demonstrated that adjunctive GAP and full-mouth SRP for the management of untreated periodontitis resulted in no remarkable improvements in the clinical parameters compared to the conventional SRP. These findings are consistent with those from previous studies (*Tsang, Corbet & Jin, 2018*; *Lelouvier et al., 2016*). Our results may be attributable to the effects of SRP, which may have relieved most of the periodontal inflammation and overshadowed the effects of the adjunctive GAP. It is noteworthy that the proportion of blood samples with increased levels of total bacteria was highest in the traditional SRP group and lowest in Group B2 (subgingival GAP right before

full-mouth SRP) the day after treatment. This result highlights the significance of the disinfection process before invasive SRP in order to reduce bacterial penetration into the bloodstream during periodontal treatment. The presence of calculus did not impair the debridement efficacy of GAP (*Petersilka, 2011*). Group B1 received adjunctive GAP right after full-mouth SRP for the purpose of joint enhanced bacteria removal and presented with *Porphyromonas* as the dominant bacterial genera in the blood the day after full-mouth SRP. The genus *Porphyromonas* is generally associated with periodontal diseases (e.g., *Porphyromonas gingivalis*) (*Pereira et al., 2017*). The results of this study indicate that the use of GAP before SRP treatment provides better safety. However, full-mouth SRP may induce a moderate acute inflammatory response, especially 24 h after treatment (*Morozumi et al., 2018*; *Graziani et al., 2015*; *Graziani et al., 2010a*; *Graziani et al., 2010b*). A similar inflammatory response was observed in the present study the day after periodontal treatment. A sharp increase in several inflammatory biomarkers, including C-reactive protein (CRP), interleukin (IL)-6, tumor necrosis factor (TNF)-α and D-dimer, may increase the vascular risk and mortality of patients. Post-operative bacteremia is the primary reason for this inflammatory response (*D'Aiuto et al., 2004*; *Graziani et al., 2010a*; *Graziani et al., 2010b*). Whether removing more oral bacteria using GAP during periodontal treatment can alleviate bacteremia following SRP requires further investigations.

There are certain limitations to the present study. Due to the nature of pilot trials, the sample size was relatively small; the study only included the descriptive analysis of blood microbiota and functional analysis was not performed. In addition, correlation analysis between periodontal parameters and the blood microbiota is lacking. Furthermore, the influence of additional GAP on the oral microbiota and the correlation between the blood and oral microbiota pre- and post- periodontal treatment were not investigated.

## CONCLUSION

The study suggests that local periodontal treatment merely influences the stability of blood microbiota for a short period. The treatment of periodontitis using adjunctive GAP right before full-mouth SRP is a promising approach to reduce the introduction of bacteria into the bloodstream during the procedure.

## ACKNOWLEDGEMENTS

We are grateful to Mrs. Shi Qiuxiang for her assistance with data management.

### Funding

The study was supported by the Qingdao Key Health Discipline Development Fund. The funders had no role in study design, data collection and analysis, decision to publish, or preparation of the manuscript.

## Grant Disclosures

The following grant information was disclosed by the authors:

Qingdao Key Health Discipline Development Fund.

## Competing Interests

The authors declare there are no competing interests.

## Author Contributions

- Wenyi Zhang performed the experiments, authored or reviewed drafts of the paper, and approved the final draft.
- Yang Meng analyzed the data, authored or reviewed drafts of the paper, and approved the final draft.
- Jin Jing analyzed the data, prepared figures and/or tables, and approved the final draft.
- Yingtao Wu and Shu Li conceived and designed the experiments, authored or reviewed drafts of the paper, and approved the final draft.

## Human Ethics

The following information was supplied relating to ethical approvals (i.e., approving body and any reference numbers):

The study protocol was approved by the Research Ethics Committee of Shandong Stomatological Hospital (No.20191003).

## Data Availability

Raw data, including clinical parameters (Probing depth, BOP, BI, PLI) at pre-and post treatment as well as the data for 16SrRNA high-throughout gene sequencing (relative OTUs taxonomy), are available in the Supplemental Files.

## Supplemental Information

Supplemental information for this article can be found online at http://dx.doi.org/10.7717/peerj.10846#supplemental-information.

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
