# Peer review of "Influence of periodontal treatment on blood microbiotas: a clinical trial"

_PeerJ, doi:10.7717/peerj.10846_

## Round 0.1 · original submission · Major Revisions

There are many issues some perhaps difficult to be handled by a revised draft.

Reviewer 1 ·

Basic reporting

Please see comments to the authors.

Experimental design

Please see comments to the authors.

Validity of the findings

Please see comments to the authors.

Additional comments

This study aimed to investigate the effects of periodontal treatment on the abundance and diversity of microbiota found in blood microbiota, before and after periodontal treatment. A total of 27 periodontitis patients were recruited and combinations of SRP and glycine air polishing (GAP) were used. Peripheral blood samples were then obtained at baseline and after treatment (6 weeks). Nested polymerase chain reaction (PCR) was used to evaluate the microbiological outcomes. Following treatment, all participants exhibited significant improvements in clinical parameters, but no significant differences were observed between treatment groups. Differences in total bacterial counts and species diversity were observed between treatment groups, whereas Porphyromonas was the dominant genera in one of the treatment groups the next day following treatment. No significant differences in relative abundances and α-diversity was observed between baseline and 6 weeks. The study concludes that local periodontal treatment merely disrupts the stability of blood microbiota in the short term, and that a combination of SRP and GAP is a promising approach to result in limited bacteraemia.
1. While the study appears to be methodologically well performed, matched data on the effect on this intervention on the oral microbiota is missing. This could have been reflected in saliva or supra/subgingival biofilms from selected sites. If such data is not available, this should at least be discussed as a limitation of the present and references should be made to other works that have investigated the effect of SRP on oral microbiota (saliva or biofilms) using next generation sequencing.
2. From the methodology and presented data it is evident that 16S rRNA Gene Sequencing was performed, using a MiSeq platform. This is not mentioned in the Abstract, and this important omission prevents the reader from understanding the experimental and analytical processes that have taken place in this work. Therefore, this should be mentioned in the Abstract.
3. The abbreviation GAP should be defined in the Abstract.
4. While there are several genera that seem to have been affected (evidenced in Figures 3 and 4), only Porphyromonas is mentioned in the Abstract. More of them should be mentioned accordingly, in order to give a better overview.
5. At repeated parts in the text, “16srRNA” is written. The correct definition is in fact “16S rRNA”, and this should be corrected accordingly.
6. Data is presented on the genus, but not species, taxonomic level (Figures 3 and 4). For a higher resolution of the presented data, it would be helpful to provide data also on the species level.

Reviewer 2 ·

Basic reporting

No comment.

Experimental design

No comment.

Validity of the findings

No comment.

Additional comments

The article meets the PeerJ criteria and should be accepted as is.

Reviewer 3 ·

Basic reporting

This study had an interesting topic, which is already widely investigated in the past, but this investigation introduced a new procedure (Air-flow), who is nowadays more and more widely used in dental practices. The question whether the use of Air-flow would translocate more or less bacteria in the blood stream is interesting, especially when it comes to treat patients with severe systemic co-morbidities where a massive translocation of bacteria may be harmful.
However, regardless the promising aim of this study, there a couple of substantial issue that make the results of this experiment of difficult interpretation.

Experimental design

- The first and most important issue is the moment when blood samples are taken. The baseline blood sampling is done at the day of the baseline visit. This is before any procedure is done. Between visit 1 and visit 2 there is no information about the time frame. The most important pitfall is that there is no blood sampling before subgingival scaling (visit 2 in figure 1). This should be the 0-time sample to really test the short term effect of the treatment. Without this is not possible to assess the real effect of the treatment both at short term and long term. Furthermore there is no blood sample taken immediately after treatment (visit 2 in Figure 1). This should be the sample to assess the difference between the groups due to the therapy. The authors took one sample one day after intervention. This is too late to assess the real effect of the treatment. There is literature that showed that already after few hours there is a clearance of bacteria in the blood stream. Furthermore, we know from previous studies that in inflamed situations brushing and even chewing play a role in the translocation of oral bacteria in the systemic bloodstream. These variables are totally not controlled and may have influenced the translocation of bacteria in the first day following the treatment. In order to be able to assess the role of the treatment, blood samples should have been collected right before and right after treatment. This is the only way to assess bacteremia due to periodontal intervention. A oral supra- and subgingival plaque sample should be also included in order to correlate the type of bacteria in the mouth with the ones found in the bloodstream.
- Because of this issue, the conclusion of this study are not appropriate because not supported by an appropriate design.
- A second issue is the use of GAP. This procedure, that should be the positive intervention is used in both B1 and B2 but in different moments. The rationale behind this choice is difficult to understand and it should be widely discussed.
- A third issue is related to the sample size. This size is clearly underpowered in order to be able to detect a clinical and statistical significant difference between the groups. This is confirmed by the authors themselves in their description of the power calculation. For microbiomic studies, multiple correction for all genera at least should be done if the aim is to identify significant difference in abundance of single genera or species. Furthermore the control group is clearly younger even if no statistical difference is found (because of the low power).

Validity of the findings

Due to the experimental issue, the validity of the findings are jeopardized. Furthermore, - there is a variation in the treatment result, but the low power was not sufficient to let detect any statistical significant difference. This variation should be used to correct for changing in the bacterial count in the blood.

Additional comments

Dear Authors, your work if for sure appreciable, but the experimental design issues make the data of this experiment difficult to interpret.
There are other issues more related to the lay-out of the manuscript listed below
- The lay out from tables and figures needs improvement. There are also an amount of abbreviations that are not explained and not consistent with the text.
- Lay-out references
- Mattila et al 1989 should be replaced by a recent reference
- The English language is sufficient but it should be deeply revised.

---

## Round 0.2 · accepted · Accept

it seems that there are no further issues raised in this round of review.

Reviewer 1 ·

Basic reporting

No comment

Experimental design

No comment

Validity of the findings

No comment

Additional comments

The authors have addressed the comments efficiently.